# The Impact of Childhood Socioeconomic Status on Adolescents’ Risk Behaviors: The Role of Physiological and Psychological Threats

**DOI:** 10.3390/ijerph192215254

**Published:** 2022-11-18

**Authors:** Xiaowei Geng, Jinrong Xu, Yicong Li, Feng Zhang, Xinye Sun, Hongyang Yu

**Affiliations:** 1Jing Hengyi School of Education, Hangzhou Normal University, Hangzhou 311121, China; 2School of Educational Science, Ludong University, Yantai 264025, China; 3Academy of Psychology and Educational Sciences, Southern Federal University, Taganrog 344006, Russia

**Keywords:** childhood socioeconomic status, adolescents, risk behavior, gain domain, loss domain

## Abstract

Adolescence is a period of high levels of risk behavior. The present research aims to examine the influences of childhood socioeconomic status (SES) on risk behaviors in gain or loss domains among adolescents and the roles of threats in this effect. In experiment 1, a total of 107 adolescents (*M_age_* = 14.80; *SD_age_* = 1.15) were asked to complete the childhood socioeconomic status scale before they took part in a risk behavior task under the gain and loss situation. A total of 149 adolescents (*M_age_* = 14.24; *SD_age_* = 1.11) in experiment 2a and 139 adolescents (*M_age_* = 13.88; *SD_age_* = 1.09) in experiment 2b completed the childhood socioeconomic status scale before they took part in a risk behavior task under the gain and loss situation under physiological threats and psychological threats, respectively. The results showed that high-childhood-SES adolescents tend to take more risks than low-childhood-SES adolescents in the gain domain, while low-childhood-SES adolescents tend to take more risks than high-childhood-SES adolescents in the loss domain. Threats amplified the impact of childhood socioeconomic status on adolescents’ risk behaviors in the gain and loss domains. When a physiological threat or psychological threat was primed, compared to the control group, in the gain situation, the extent to which high-childhood-SES adolescents showed greater risk seeking than low-childhood-SES adolescents became larger; in the loss domain, the extent to which low-childhood-SES adolescents showed greater risk seeking than high-childhood-SES adolescents became larger.

## 1. Introduction

Adolescence generally refers to the adolescent developmental period between the ages of 11/12 and 18/19, during which individuals rapidly develop physically and mentally, thus exhibiting characteristics such as emotional sensitivity, instability, and poor self-control, and it is a period of high risk seeking [1]. Most previous studies on adolescents’ risk behaviors have focused on negative risk behaviors (e.g., stealing, smoking, driving under the influence of alcohol, etc.). These studies have found that factors influencing adolescent risk behaviors include family functioning [2], parent–child relationships [3], peer influence [4], sensation seeking [5], and socioeconomic status [6,7]. Few studies have shed light on the correlation between childhood socioeconomic status and adolescents’ risk behaviors, so it is unclear how childhood socioeconomic status affects adolescents’ risk behaviors.

Childhood socioeconomic status (SES) refers to the extent to which an individual grew up in resource-plentiful vs. resource-scarce environment [8], which is one kind of socioeconomic status [9]. Griskevicius et al. (2011) argued that childhood was a critical or sensitive period for the formation of life history strategies and that, compared with current socioeconomic status, childhood SES has a greater impact on individuals’ life history strategies. Research has suggested that the effect of stress on health was influenced by people’s childhood SES rather than their current SES. Childhood SES is an important indicator of childhood poverty and has a more profound impact on individuals’ psychological well-being than adulthood SES. It not only determines the material resources that individuals possess, but also determines the development of individual psychological levels [10]. Therefore, it is necessary to study how childhood SES influences adolescents’ risk-seeking behaviors.

In many real-world risk decisions, decisions could be in the domain of gain or in the domain of loss. In the gain domain, the goal of people is to maximize gains, while in the loss domain, the goal is to minimize losses. Prospect theory proposes that individuals are more prone to risk seeking in loss domains and more conservative in gain domains [11]. Based on prospect theory, many studies have found the same results [12,13]. However, few studies shed light on adolescents’ risk behaviors of gain and loss domain. In addition, previous studies showed that individuals with different SES were differently sensitive to the environment, with low-SES individuals being more sensitive to threats [14]. It is not clear whether both physical and psychological threats moderate the influence of childhood SES on adolescents’ risk behaviors. Therefore, the present study aims to examine the influences of childhood SES on risk behaviors in gain or loss domains among adolescents and the roles of threats in this effect.

### 1.1. Childhood SES and Adolescent Risk Behaviors in Gain and Loss Domains

Previous research found that childhood SES had a greater effect on life history strategies than current socioeconomic status [8]. Life history strategies are how individuals allocate their limited resources to achieve their ultimate functional goals, including survival and reproduction in different environments. People with slow life history strategies tend to seek future survival investments and long-term benefits, such as late marriage and late childbirth and delayed gratification behaviors; fast life history strategies prioritize present investment and place more emphasis on current benefits. Griskevicius et al. (2013) found that individuals with childhood financial insecurity were more inclined to adopt fast life history strategies, which was associated with risk taking and prioritizing present benefits; in contrast, those with childhood financial security were more inclined to adopt slow life history strategies, showing less impulsivity and prioritizing future benefits. Risks are usually associated with waiting for future long-term benefits, such as the possibility of unexpected death before receiving future benefits. Low-childhood-SES individuals might not afford to take the risk of gaining nothing whilst waiting for greater future benefits. In contrast, high-childhood-SES individuals could afford to take the risks of gaining nothing during the waiting for future larger benefits. Therefore, high-childhood-SES adolescents tend to take more risks than low-childhood-SES individuals in gain domains.

According to the uncertainty management theory, which suggests that early life deprivation leads to a set of preferences aimed at minimizing the downside costs of uncertainty across a variety of domains [15], low-childhood-SES individuals focus primarily on reducing losses in uncertain circumstances, exhibiting more loss aversion. As a result, low-childhood-SES individuals are more inclined to exhibit risk-seeking behavior in loss domains in order to avoid loss. As a result, the following hypothesis is proposed:

**H1a:** *In gain domains, high-childhood-SES adolescents show more risk seeking than those with low-childhood-SES*;

**H1b:** *In loss domains, low-childhood-SES adolescents show more risk seeking than those with high-childhood-SES adolescents*.

### 1.2. The Moderating Role of Threats

Threats are pervasive negative stimuli that can trigger negative emotions such as fear, anxiety, and tension and have an impact on people’s survival and development [16]. It has been shown that individuals who grow up in harsh and unpredictable environments are more likely to perceive threats as uncontrollable and feel unable to protect themselves from such uncertainty. In contrast, individuals who grow up in less harsh and predictable environments are more likely to perceive threats as controllable and believe that they can protect themselves from this uncertainty by changing their behaviors [17]. Therefore, it can be argued that threats have a greater impact on low-childhood-SES individuals and a smaller impact on those with high childhood SES. Furthermore, one study found that low-childhood-SES individuals more frequently develop over-eating behaviors after priming death threats [18]. A study by Geng, Zhang, and Liu (2022) [19] also found that, for college students with high subjective socioeconomic status, public crises do not significantly influence their intertemporal decision making; for those with low subjective socioeconomic status, public crises led to a greater preference for smaller-sooner benefits. Belsky et al. [20] found that, compared to low-childhood-SES individuals, high-childhood-SES individuals were more likely to make optimal strategies, tending towards long-term benefits and delayed gratification after priming the death threats or negative life circumstances. Therefore, we hypothesize that threats may amplify the effect of childhood SES on adolescents’ risk behaviors in gain and loss domains. That is to say, threat amplifies the difference between the risk behaviors of adolescents with high childhood SES and low childhood SES. Specifically, under the condition of threat priming, in the gain domain, the extent to which adolescents with high childhood SES showed greater risk seeking than those with low childhood SES would become larger than the control condition. In contrast, under the condition of threat priming, in the loss domain, the extent to which adolescents with low childhood SES show greater risk seeking than those with high childhood SES would become larger than the control condition. The following hypothesis is proposed:

**H2:** *Threats amplify the effect of childhood SES on adolescents’ risk-seeking behaviors in the gain and loss domains*.

In order to examine these hypotheses, three experiments were designed in the present research. Specifically, experiment 1 examined the effect of childhood SES on adolescents’ risk seeking behaviors in the gain and loss domains, while experiments 2a and 2b examined the moderating role of physiological threat and psychological threat in the effect of childhood SES on adolescents’ risk behaviors in the gain or loss domain, respectively.

## 2. Experiment 1: The Effect of Childhood SES on Adolescents’ Risk-Seeking Behaviors in Gain and Loss Domains

### 2.1. Participants

Based on G*Power, it was calculated that a total of 98 participants were required for the two-factor, mixed-design, repeated-measures ANOVA (1 − *β* = 0.8, *α* = 0.05, *f* = 0.25). In total, 107 junior and senior high school students (38 males) were recruited from a city in China. The mean age of participants was 14.80 years (*SD* = 1.15).

### 2.2. Experimental Design

We adopted a 2 × 2 mixed experimental design, with a between-subject variable (childhood SES: high vs. low) and a within-subject variable (gain domain vs. loss domain). The dependent variable was risk-seeking behavior.

### 2.3. Materials

#### 2.3.1. Childhood Socioeconomic Status Scale (Childhood SES)

Adolescents’ childhood socioeconomic statuses were assessed using a 4-item scale [8,21]. Participants rated the items (e.g., “I grew up in a relatively wealthy neighborhood” and “My family usually had enough money for things when I was growing up”) using a 7-point scale (1 = strongly disagree, 7 = strongly agree). The mean score of childhood SES in this study was 4.40 (*SD* = 1.05). The average was converted into a Z-score. A positive Z-score was coded as high childhood SES and negative Z-score was coded as low childhood SES.

#### 2.3.2. Risk-Seeking Measurement in the Gain and Loss Domains

The previous research on risk decision making mostly focuses on the economic domain [8,15,21]. In order to improve the ecological validity of the risk-seeking behavior, following Liu et al. (2010) [22], the risk-seeking measurement involved three domains, namely health, economic and recreation, with each containing three questions, leading to a total of nine questions. The risk-seeking behavior questions in the gain domain have two alternatives, one for the less risky situation and the other for the more risky situation: e.g., A. 80% receive USD 10; B. 40% receive USD 20, with A coded as 0 and B coded as 1. The risk-seeking behavior question in the loss domain also has two alternatives: for example, C. 80% lose USD 10 and D. 40% lose USD 20, with C coded as 0 and D coded as 1. Thus, there were 9 items for risk-seeking assessment in the gain domain and loss domain. A total score of the nine choices (the lowest score is 0 and the highest score is 9) was used as the index of risk seeking in the gain domain and loss domain, respectively, with higher scores indicating a greater propensity for risk seeking. The internal consistent reliability coefficient of the risk-seeking measurement was 0.70 in the present research, indicating high reliability.

### 2.4. Procedures

The questionnaires were completed through E-prime on the computer. First, participants provided information on their childhood SES. Then, participants completed the risk-seeking tasks in both the gain and loss domains. The order of the gain domain and loss domain was counterbalanced.

### 2.5. Data Analyses

All questionnaire data were processed and analyzed by SPSS (the Statistical Package for the Social Sciences), using statistical methods, such as the independent sample *t*-test, ANOVA, and a simple effect analysis.

### 2.6. Results

Results of a 2 (childhood SES: high vs. low) × 2 (gain domain vs. loss domain) repeated-measures ANOVA revealed that the effect of the gain or loss domain was significant, *F*(1, 105) = 45.16, *p* < 0.001, *η*^2^ = 0.30, and that adolescents in the loss domain (*M* = 5.74; *SD* = 2.03) were more risk seeking than those in the gain domain (*M* = 4.14; *SD* = 2.32); the effect of childhood SES was not significant, *F*(2, 104) = 1.00, *p* = 0.32, *η*^2^ = 0.01. The interaction effect of the gain or loss situation and childhood SES was highly significant, *F*(1, 105) = 20.54, *p* < 0.001, *η*^2^ = 0.16; see Figure 1. A simple effects analysis found that, in the gain domain, compared to low-childhood-SES adolescents (*M_low SES_* = 3.35; *SD_low SES_* = 1.88), high-childhood-SES adolescents (*M_high SES_* = 4.81; *SD_high SES_* = 2.45) were more risk seeking. However, in the loss domain, low-childhood-SES individuals (*M_low SES_* = 6.18; *SD_low SES_* = 2.06) were more risk seeking than high-childhood-SES individuals (*M_high SES_* = 5.36; *SD_high SES_* = 1.94), *F*(1, 105) = 4.52, *p* = 0.04, *η*^2^ = 0.36.

### 2.7. Discussion

Experiment 1 found that high-childhood-SES adolescents tend to take more risks than low-childhood-SES adolescents in the gain domain, while low-childhood-SES adolescents tend to take more risks than high-childhood-SES adolescents in the loss domain. The findings were consistent with H1.

Experiment 1 only investigated how childhood SES influenced risk preferences in the gain and loss domains. Experiment 2 will further examine the effects of childhood SES on adolescents’ risk-seeking behaviors in the gain and loss domains under physiological and psychological threats. Physiological threats are threats that violate an individual’s body or are in conflict with their physiological needs, including a lack of food, physical injury, and sudden loud noises [23]. Psychological threats are information that threaten the interests of individuals or the groups they belong to during social interactions [24], such as social rejection, social exclusion, self-threat, and relationship threats. We will examine the moderating effects of physiological threats and psychological threats through Experiment 2a and Experiment 2b, respectively.

## 3. Experiment 2: The Moderating Role of Threats

### 3.1. Experiment 2a: The Moderating Role of Physiological Threat

#### 3.1.1. Participants

Based on G*Power, it can be concluded that a total of 136 participants were required for the repeated-measures ANOVA of the three-factor mixed design (1 − *β* = 0.8, *α* = 0.05, *f* = 0.25). The participants were 149 middle and high school students (68 males, 81 females) from a city in China. The mean age of participants was 14.24 years (*SD* = 1.11).

#### 3.1.2. Experimental Design

We adopted a 2 (childhood SES: high vs. low) × 2 (gain domain vs. loss domain) × 2 (physiological threat priming vs. control condition) mixed experimental design, with childhood SES and physiological threat priming as between-subject variables, and gain/loss as a within-subject variable. The dependent variable was risk-seeking behavior.

#### 3.1.3. Materials

(1) Physiological threat priming materials

The Geneva affective picture database (GAPED) was used to prime physiological threat [25], which included eight specific threatening pictures such as spiders and snakes. In addition, non-threatening pictures such as scenes and tools were used as control conditions. To test whether the priming of physiological threat was effective, 107 adolescents were recruited. The mean age of the 29 males was 15.97 years (*SD* = 0.61) and the mean age of the 78 females was 15.99 years (*SD* = 0.34). Additionally, they were randomly assigned to one of the two conditions: physiological threat priming condition and control condition. After this, participants were asked to assess the extent to which they felt threat on a 5-point scale. An independent sample T-test found that, compared to the control group (*M* = 1.31; *SD* = 0.67), the physiological threat priming group felt threat to a greater extent (*M* = 3.21; *SD* = 1.06), *t* (87.28) = 11.01, *p* < 0.001, 95%*CI =* [1.55, 2.23], *d* = 2.14, indicating that physiological threat priming was effective.

(2) The childhood SES measure and the risk-seeking behavior task in the gain and loss domains were the same to that of experiment 1.

#### 3.1.4. Procedure

The questionnaire was completed through E-prime on the computer. First, participants provided information on their childhood SES. Then, participants received physiological threat priming. Finally, participants completed the risk-seeking behavior tasks both in the gain and loss domains.

#### 3.1.5. Data Analyses

All questionnaire data were processed and analyzed by SPSS (the Statistical Package for the Social Sciences), using statistical methods, such as the independent sample *t*-test, ANOVA, and the simple effect analysis.

#### 3.1.6. Results

The results of a repeated-measures ANOVA of 2 (physiological threat priming vs. control condition) × 2 (childhood SES: high vs. low)× 2 (gain or loss domains: gain vs. loss) revealed that the effect of the gain or loss domain was significant, *F*(1, 145) = 72.08, *p* < 0.001, *η*^2^ = 0.33, with adolescents being more risk seeking in the loss domain (*M* = 5.09, *SD* = 2.19) than in the gain domain (*M* = 3.82; *SD* = 2.21); the effect of physiological threat priming was non-significant, *F*(2, 144) = 1.33, *p* = 0.25, *η*^2^ = 0.01; the main effect of childhood SES was not significant, *F*(2, 144) = 0.07, *p* = 0.80, *η*^2^ = 0.01; the interactive effect of the gain or loss domain, threat priming and childhood SES was significant, *F*(1, 145) = 7.67, *p* = 0.01, *η*^2^ = 0.05.

Further simple effect analysis showed that, in the control condition, the interaction effect of the gain or loss domain and childhood SES was significant, *F*(1, 73) = 24.60, *p* < 0.001, *η*^2^ = 0.25). As shown in Figure 2, in the gain domain, high-childhood-SES adolescents (*M_high SES_* = 4.20; *SD_high SES_* = 2.28) were more risk seeking than those with low childhood SES (*M_low SES_* = 3.34; *SD_low SES_* = 1.97), but the difference was not significant, *t*(73) = −1.73, *p* = 0.09, 95%*CI =* [−1.84,0.13], *d* = 0.40; in the loss domain, low-childhood-SES adolescents (*M_low SES_* = 5.46; *SD_low SES_* = 2.13) were more risk seeking than high-childhood-SES adolescents (*M_high SES_* = 4.15; *SD_high SES_* = 1.15)), *t*(73) = 2.89, *p* = 0.01, 95%*CI =* [0.40, 2.21], *d* = 0.67. When physiological threat was primed, the interaction effect of the gain or loss domain and childhood SES was highly significant, *F*(1, 72) = 54.26, *p* < 0.001, *η*^2^ = 0.43. As shown in Figure 2, in the gain domain, high-childhood-SES adolescents (*M_high SES_* = 4.70; *SD_high SES_* = 2.27) were more risk seeking than low-childhood-SES adolescents (*M_low SES_* = 2.57; *SD_low SES_* = 1.55), *t*(72) = −4.50, *p* < 0.001, 95%*CI =* [−3.09, −1.19], *d* = 1.10; in the loss domain, low-childhood-SES adolescents (*M_low SES_* = 6.60; *SD_low SES_* = 1.67) were more risk seeking than high-childhood-SES adolescents (*M_high SES_* = 4.61; *SD_high SES_* = 2.30), *t*(72) = 4.05, *p* < 0.001, 95%*CI =* [1.01, 2.97], *d* = 0.99. In other words, physiological threat amplified the effect of childhood SES and the gain/loss domain on adolescents’ risk behaviors.

### 3.2. Experiment 2b: The Moderating Effect of Psychological Threats

#### 3.2.1. Participants

Based on G*Power, it can be concluded that a total of 136 participants were required for the repeated-measures ANOVA of the three-factor mixed design (1 − β = 0.8, α = 0.05, *f* = 0.25). The participants were 139 middle and high school students (66 males, 73 females) from a city in China. The mean age of participants was 13.88 years (*SD* = 1.09).

#### 3.2.2. Experimental Design

We adopted a 2 (childhood SES: high vs. low) × 2 (gain domain vs. loss domain) × 2 (psychological threat priming vs. control condition) mixed experimental design, with childhood SES and psychological threat priming as between-subject variables, and gain/loss as within-subject variables. The dependent variable was risk-seeking behavior.

#### 3.2.3. Materials

(1) Psychological threat priming materials

Following Griskevicius et al. (2011) [8], participants were asked to read one of two news articles. One was titled Why the Generation Z are called the “Beat Generation”, which was very critical on Generation Z, the other was titled The Generation Z represents the younger generation, which was a comprehensive and objective evaluation of Generation Z. Participants were told that the two articles were selected from the China Youth Network, which is an influential and credible platform. Both articles had similar lengths and structures (approximately 700 words). To test the effectiveness of psychological threat priming, 136 junior high school students (63 males and 73 females; *M_age_* = 13.49; *SD_age_* = 0.68) were recruited and randomly assigned into a psychological threat priming group and a control group. Participants were asked to rate the perceived sense of threat immediately after reading the news article (1 = not at all, 5 = very strongly). An independent samples t-test found that the psychological threat priming group (*M* = 3.12; *SD* = 1.14) experienced a stronger sense of threat than the control group (*M* = 1.93; *SD* = 0.78), indicating that psychological threat priming was effective, *t* (118.35) = 7.12, *p* < 0.001, 95%*CI* [0.86, 1.52], *d* = 1.22.

(2) The results from the Childhood SES scale and the risk-seeking behavior task in the gain and loss domains were the same as they were in experiment 1.

#### 3.2.4. Procedure

The questionnaires were completed through E-prime on the computer. First, participants completed the childhood SES measurement. Then, participants received psychological threat priming. Finally, participants completed the risk-seeking behavior tasks both in the gain and loss domains.

#### 3.2.5. Data Analyses

All questionnaire data were processed and analyzed by SPSS (the Statistical Package for the Social Sciences), using statistical methods, such as the independent sample *t*-test, ANOVA, the simple effect analysis.

#### 3.2.6. Results

The results of a repeated-measures ANOVA of 2 (psychological threat priming vs. control condition) × 2 (childhood SES: high vs. low) × 2 (gain or loss domains: gain vs. loss) showed that the main effect in the gain or loss domain was significant, *F*(1, 135) = 25.10, *p* < 0.001, *η²* = 0.16, with adolescents being more adventurous in the loss situation (*M* = 5.10, *SD* = 2.23) than in the gain situation (*M* = 4.19; *SD* = 2.47); the main effect of threat priming was not significant, *F*(2, 134) = 0.03, *p* = 0.87, *η²* = 0.01; the main effect of childhood SES was not significant, *F*(2, 134) = 0.34, *p* = 0.56, *η²* = 0.01; the interactive effect of the gain or loss domain, threat priming and childhood SES was significant, *F*(1, 135) = 6.53, *p* = 0.01, *η²* = 0.05.

Further simple analysis showed that, in the control condition, the interaction effect between the gain/loss domain and childhood SES was significant, *F*(1, 59) = 9.52, *p* = 0.003, *η²* = 0.14. As shown in Figure 3, in the gain domain, high-childhood-SES adolescents (*M_high SES_* = 4.76; *SD_high SES_* = 3.30) were more risk seeking than low-childhood-SES adolescents (*M_low SES_* = 4.13; *SD_low SES_* = 1.68), but the difference was not significant, *t*(56.36) = −0.98, *p* = 0.33, 95%*CI* [−1.92,0.65], *d* = 0.24; in the loss situation, low-childhood-SES adolescents (*M_low SES_* = 5.17; *SD_low SES_* = 1.49) were more risk-averse than high-childhood-SES adolescents (*M_high SES_* = 4.68; *SD_high SES_* = 2.93), but the difference was not significant, *t*(56.42) = 0.86, *p* = 0.39, 95%*CI* [−0.65,1.63], *d* = 0.21. When psychological threat was primed, the interactive effect of the gain or loss domain and childhood SES was highly significant (*F*(1, 76) = 28.96, *p* < 0.001, *η²* = 0.28). As shown in Figure 3, in the gain domain, high-childhood-SES adolescents (*M_high SES_* = 4.89; *SD_high SES_* = 1.95) were more risky than those low-childhood-SES adolescents (*M_low SES_* = 3.10, *SD_low SES_* = 2.05), *t*(76) = −3.95, *p* < 0.001, 95%*CI* [−2.70,−0.89], *d* = 0.89; in the loss domain, low-childhood-SES adolescents (*M_low SES_* = 5.80, *SD_low SES_* = 1.83) were more risk seeking than high-childhood-SES adolescents (*M_high SES_* = 4.70; *SD_high SES_* = 2.11), *t*(76) = 2.47, *p* = 0.02, 95%*CI* [0.21,1.99], *d* = 0.56. In other words, psychological threat positively moderated the effect of childhood SES and the gain/loss domain on adolescents’ risk behaviors.

## 4. General Discussion

### 4.1. Influences of Childhood SES on Adolescents’ Risk Behaviors in Gain and Loss Domains

Experiment 1 found that childhood SES significantly influenced adolescents’ risk behaviors in the gain and loss domains. In the gain domain, high-childhood-SES adolescents were more risk seeking than those with low childhood SES. However, in the loss domain, low-childhood-SES adolescents were more risk seeking than those with high childhood SES, which was consistent with hypothesis 1. High-childhood-SES adolescents usually have more resources and can afford to take risks; therefore, they tend to take more risks to achieve gains. Previous studies have found that high-childhood-SES individuals are more inclined to adopt low life history strategies and prioritize long-term benefits, which are associated with high risk [8,21]. Therefore, individuals with high childhood SES tend to be risk seeking in order to receive greater benefits. In contrast, low-childhood-SES adolescents usually experienced early-life deprivation and have fewer resources to allow risk-seeking behavior. Previous studies have found that low-childhood-SES individuals are more inclined to adopt fast life history strategies and prioritize current benefits [8,21]. According to the uncertainty management theory [15], early-life deprivation leads low-childhood-SES individuals to focus on avoiding losses associated with uncertain environment and showing loss aversion. Therefore, in the loss domain, low-childhood-SES adolescents were more willing to exhibit risk-seeking behaviors in order to avoid losses than high-childhood-SES individuals.

### 4.2. Influences of Childhood SES on Adolescent Risk-Seeking Behavior in Gain and Loss Domains under Threat

Experiment 2 found that physiological and psychological threats amplified the effect of childhood SES on adolescents’ risk-seeking behaviors in the gain and loss domains. Compared to the control group, when threats were primed, in the gain situation, the extent to which high-childhood-SES adolescents showed greater risk seeking than low-childhood-SES adolescents became larger; in the loss domain, the extent to which low-childhood-SES adolescents showed greater risk seeking than high-childhood-SES adolescents became larger. Low-childhood-SES individuals are more likely to perceive threats as uncontrollable and believe that they cannot protect themselves from such uncertainty. Conversely, high-childhood-SES individuals are more likely to perceive threat as controllable and believe that they can protect themselves from this uncertainty by changing their behaviors [17]. Therefore, threats can intensify the loss aversion of low-childhood-SES adolescents and thus they become more risk seeking in loss domains to avoid losses. For the same reason, threats also intensify the risk-averse tendencies of low-childhood-SES adolescents in the gain domain. Thus, physiological and psychological threats amplify the effect of childhood SES on adolescents’ risk-seeking behaviors in the gain and loss domains.

### 4.3. Theoretical and Practical Value

Previous research has mostly focused on negative risk-seeking behaviors, while less research has investigated the risk behaviors of adolescents with high (versus low) childhood SES in gain or loss domains. This study investigated the influence of childhood SES on adolescents’ risk behaviors in the gain and loss domains and found that high-childhood-SES adolescents tend to be more risk seeking than low-childhood-SES adolescents in the gain domain, while low-childhood-SES adolescents tend to be more risk seeking than high-childhood-SES adolescents in the loss domain, which provides new insights of how childhood SES affects adolescents’ risk behaviors. In addition, this study examines the role of both physiological and psychological threats in the effect of childhood SES on adolescents’ risk behaviors in the gain and loss domains and found that threats amplify the effect of childhood SES on adolescents’ risk-seeking behaviors in the gain and loss domains, which provides a new research direction for the study of adolescent risk behavior.

The present study also has important practical implications. The findings demonstrate the risk-seeking behavior of adolescents with different childhood SES in the gain and loss domains, which can provide theoretical guidance for the intervention and guidance of adolescents’ risk behaviors. For example, our findings showed that high-childhood-SES adolescents tend to be more risk seeking than low-childhood-SES adolescents in the gain domain, while low-childhood-SES adolescents tend to be more risk seeking than high-childhood-SES adolescents in the loss domain. Based on this, teachers and parents should encourage low-childhood-SES adolescents to take more risks when achieving some gains. In contrast, when facing some losses, teachers and parents should encourage low-childhood-SES adolescents to avoid more risk to keep them safe.

### 4.4. Limitation and Directions of Future Research

There were some limitations in the present research. First, in the present study, adolescents’ risk behaviors were measured by questionnaires. In order to improve ecological validity, future research could use field studies to further examine the impact of childhood SES on adolescents’ risk-seeking behaviors in the gain and loss domains. Second, the present research used a cross-sectional design. Cross-sectional studies analyze data at a specific point in time, which has limitations for exploring the relationships between variables. The longitudinal research design can be used in future studies to provide insights into the developmental progression of individuals’ risk behaviors from childhood to adulthood.

## 5. Conclusions

The present research found that high-childhood-SES adolescents tend to be more risk seeking than low-childhood-SES adolescents in the gain domain, while low-childhood-SES adolescents tend to be more risk seeking than high-childhood-SES adolescents in the loss domain. Both physiological and psychological threats amplified the effect of childhood SES on adolescents’ risk-seeking behaviors in the gain and loss domains. When threats were primed, compared to the control group, in the gain situation, the extent to which high-childhood-SES adolescents showed greater risk seeking than low-childhood-SES adolescents became larger; in the loss domain, the extent to which low-childhood-SES adolescents showed greater risk seeking than high-childhood-SES adolescents became larger. This can provide helpful suggestions for teachers and parents to guide high-childhood-SES (vs. low) adolescents’ risk behaviors when facing gains or losses.

## Figures and Tables

**Figure 1 ijerph-19-15254-f001:**
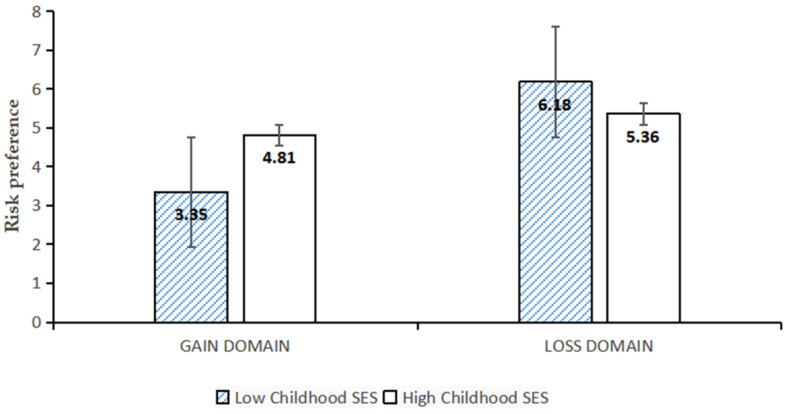
The effect of childhood SES on adolescents’ risk behaviors in gain and loss domains.

**Figure 2 ijerph-19-15254-f002:**
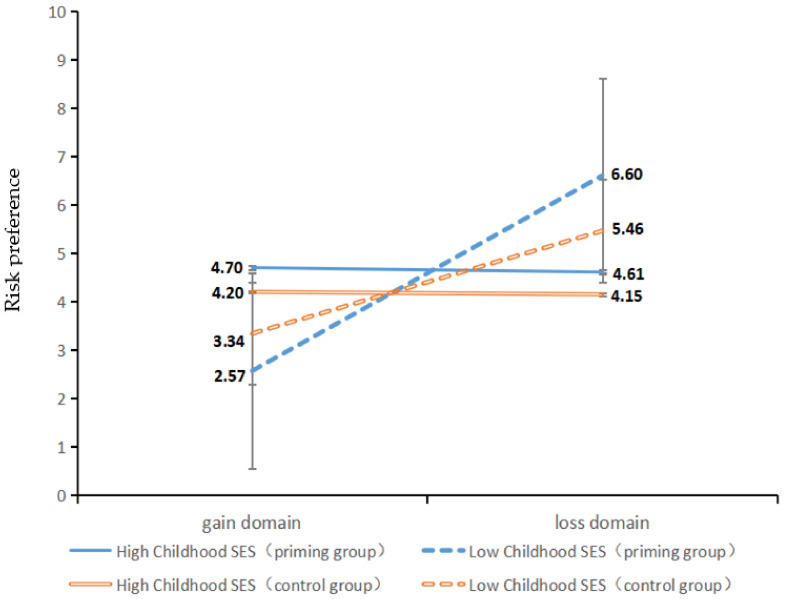
The effect of childhood SES on adolescents’ risk behaviors in the gain and loss domains under physiological threat priming.

**Figure 3 ijerph-19-15254-f003:**
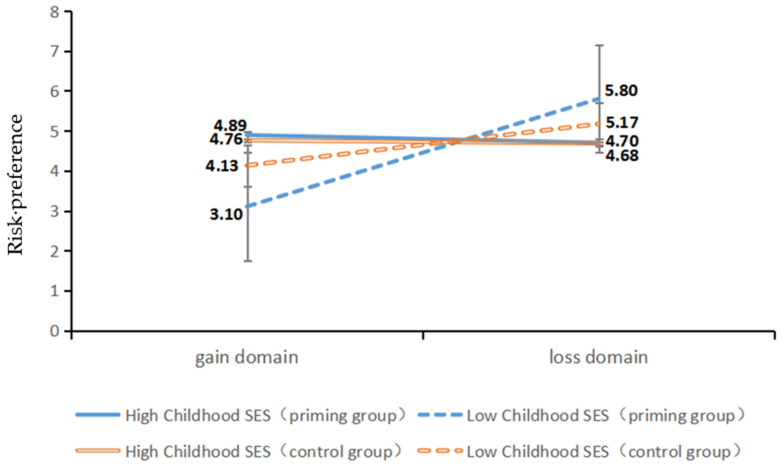
The effect of childhood SES on adolescents’ risk behaviors in the gain and loss domains under psychological threat priming.

## Data Availability

According to the data access policies, the data used to support the findings of this study are available upon reasonable request to: fengandwei@126.com.

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
