# Peer review of "The Impact of Childhood Socioeconomic Status on Adolescents’ Risk Behaviors: The Role of Physiological and Psychological Threats"

_ijerph, 2022, doi:10.3390/ijerph192215254_

Round 1

Reviewer 1 Report

The present study aimed to examine the influences of children's SES on risk behaviors in the domains of gain or loss among adolescents and the roles of threats in this effect.

In general, I consider the topic extremely relevant. After careful reading of the text, at various times he was forced to read again to build a better understanding of what the authors were looking for. I believe that it is necessary to reformulate both the contents of the Introduction and the Methodology section, which failed to convey what was done, and mainly, in what way. Here are my considerations, I hope they can help the authors!

Summary

1. inform the mean age value (SD)

2. Inform the name of the instruments used for data collection;

3. What was presented in the abstract is not the purpose of the study. Therefore, this information does not match what was presented at the end of the Introduction.

*I suggest a classic presentation with objectives, methodology, results, conclusion. It is not necessary to include these titles in the abstract, however, it is important that the reader receives this information;

**In the present form, 80% of the abstract is composed of results.

Introduction

Line 46-47: The authors introduced the following sentence "Previous studies have investigated the risk behavior of adolescents in the domain. However, they did not report the studies that stated the facts presented. Please present these studies

Line 47: right after that, the following sentence was introduced "Prospect theory proposes that individuals are more likely to seek risk in the loss domains and more conservative in the gain domains [11]." This sentence has no connection with the ideas of the previous sentence. I suggest severely reviewing this whole articulation of ideas because here is the justification for carrying out the present study. So far this is weak!

Line 49: "We have no idea whether childhood SES influences adolescents' risk behaviors in the gain and loss domains." There is no properly articulated theoretical framework for this statement. This sounds like a doubt of the authors, and not a necessity of the scientific community. Please substantiate this based on previous studies.

***Also the last paragraph must be reflected and rewritten. To date, the basis for carrying out this study is not clear!

Line 83: The continuity of the hypothesis (H1), that is, the sentence "In loss domains, low childhood SES adolescents are more

risk seeking than those with high childhood SES adolescents", seems to me to be a possible "H2". I understand that this "loss" is already another assumption. Suggestion for reflection by the authors!

Line 106: This paragraph needs better wording. The presentation of the hypothesis followed by a long explanation ends up confusing the reader. I suggest adopting a style that introduces the content, and then ends with the hypothesis.

Method

Line 134: Regarding the "Risk-Seeking tasks in the gain and loss domains" questionnaire, I have doubts:

1. If 03 questions in each of the 03 domains would be enough to examine what the study proposes;

2. I also wonder if a 14 year old average young person has the ability to respond to their family's financial issues. I believe your parents could report this more accurately!

*About the experiments

1. In general, scientific articles have a main section that presents ALL the statistical procedures. Therefore, this section is presented separately from the Results section. In the present study, there is a statistical section for each experiment, which is confused with the results. This makes it difficult to understand events;

2. How many groups were there in each of the experiments? Why use the ANOVA test? Were there more than two groups? This is not clear in Figure 1.

General Discussion and Conclusion

They were presented in a very summarized way. There could and should be greater articulation in the Discussion section with the existing literature.

Reviewer 2 Report

This is a fairly complete research paper with a good literature review and research design. I greatly appreciate the authors’ efforts and their good results on adolescents’ risk behavior. Several revisions are humbly suggested as follows.

1.Line 76: The author had cited "individual uncertainty management theory" as the theoretical of this research. The author can briefly explain the meaning of this theory, so that readers can easily understand its meaning.

2.Line 164: The caption for Figure 1. should be placed below the figure.

3.The research results are diverse and contributing. Line 342: In the "Theoretical and practical value" section, it is suggested that the authors can interpret the research findings more, so that the value and contribution of this research can be more evident.

4.Line 354: The authors explain the limitations of the current study. It is suggested that the author can write a separate paragraph on "Limitations and Directions of Future Research", and make further suggestions on research objects, sampling methods, research methods and research variables.

Reviewer 3 Report

Author(s) provides original results of their investigations and examination of material from their own collections. The applied methods and the interpretation and presentation of results correspond to international standards. Findings are useful for educational, pedagogical, social environments. Author(s) have studied and used an appropriate number of bibliography sources. The language is not always perfect, the syntax is in some parts a bit convoluted.

Required changes:

·        In my opinion only one chapter need to be correct. Conclusions need to be developed. The widely described results and the discussion of the results allow drawing more detailed conclusions.

·    I suggest to add short implications of findings and short recommendation after conclusions. 

Round 2

Reviewer 1 Report

Congratulations to the authors, the current version of the manuscript has been considerably improved.

*A revision of the sentence appears on line 58, page 2: the word "decision" was written twice in a row!